# Knowledge, attitude, and practice toward coronary heart disease secondary prevention among coronary heart disease patients in Shanghai, China

**Hao Wang** ⓘ ◎ *, **Bo Wu** ◎, **Wenqi Guan, Tan Zhou, Hongbao Wang, Wei Li, Xueqin He**

Department of Cardiovascular, YangPu Hospital, School of Medicine, Tongji University, Shanghai, China

◎ These authors contributed equally to this work.
* dr_wanghao@163.com

**Data Availability Statement:** All relevant data are within the paper and its Supporting information files.

## Abstract

### Background

This study aimed to investigate knowledge, attitude, and practice (KAP) toward coronary heart disease (CHD) secondary prevention among CHD patients.

### Methods

This web-based cross-sectional study enrolled patients with CHD who visited the Yangpu District Central Hospital in Shanghai (China) between October 18, 2022, and March 25, 2023. The administered questionnaire assessed demographic information and KAP; factors associated with good practice were identified by multivariate logistic regression.

### Results

A total of 507 participants were included in the study, with 361 (71.2%) being male. In terms of education, 125 (24.7%) had a junior high school level or below. The mean scores for knowledge, attitudes, and practices were 31.28 ± 7.30 (possible range: 0–42), 54.09 ± 3.33 (possible range: 12–60), and 35.48 ± 3.36 (possible range: 11–55), respectively. For specific knowledge items on CHD, 57.6% of participants correctly identified that women are more susceptible to CHD. Physical labor and emotional excitement as triggers for CHD were correctly recognized by 94.1%. The need for long-term medication and follow-up after a CHD diagnosis had the highest correctness rate at 98.8%. Additionally, 84.6% correctly understood that recurrence of CHD is possible after PCI surgery. Multivariate analysis indicated that smoking and diabetes status were significantly associated with Practice scores. Current smokers reported lower practice levels than never smokers (OR = 2.858, 95% CI: 1.442–5.662, P = 0.003). Participants with diabetes reported higher practice levels than those without diabetes (OR = 4.169, 95% CI: 2.329–7.463, P < 0.001).

**Funding:** The author(s) received no specific funding for this work.

**Competing interests:** The authors have declared that no competing interests exist.

**Abbreviations:** CHD, coronary heart disease; KAP, knowledge, attitude, and practice.

## Conclusions

Patients with CHD in Shanghai, China, demonstrated good knowledge and positive attitudes toward CHD secondary prevention, although there were some gaps in actual practice behaviors. Enhancing targeted educational interventions and support systems in clinical settings may help bridge these gaps and improve adherence to recommended preventive practices.

## Background

Coronary heart disease (CHD) is atherosclerosis caused heart damage and is one of the most common chronic illnesses and the most prevalent cardiovascular disease worldwide [1, 2]. CHD mortality accounts for $> 80\%$ of global cardiovascular deaths and 15.5% of total deaths annually [3, 4]. Moreover, CHD is one of the leading causes of disability worldwide, and with the ever-younger onset, the working population is becoming increasingly more affected, resulting in severe economic and social consequences [2, 5]. The primary risk factors for CHD include hypertension, high low-density lipoprotein cholesterol, smoking, diabetes mellitus, obesity, physical inactivity, unhealthy diet, and systemic inflammation. Additional factors such as genetic predisposition and vascular endothelial dysfunction also contribute to its development [1, 6, 7]. However, influencing those factors to lower the lifetime risk of CHD is challenging and thus remains the main research direction [8].

Pathogenesis of CHD is mostly well understood, with percutaneous coronary intervention (PCI) being widely used to relieve pain and improve quality of life [9]. Numerous lifestyle correction programs have been designed to target smoking habits, body weight, hypertension, and high cholesterol levels with reported success [8]. Aerobic exercises can lower elevated blood lipoproteins, affecting HDL-C maturation, transport, and catabolism [10, 11]. Although cardiac rehabilitation could significantly reduce CHD mortality and morbidity, this depends on the patient's medication compliance and willingness to fully adhere to the necessary lifestyle changes [8, 11]. Therefore, patient education is necessary to improve treatment-related attitudes and promote health behavior changes [12].

The knowledge, attitude, and practice (KAP) survey is a well-known tool used to assess patient's compliance behaviors and influence their knowledge, attitudes, and personal habits regarding CHD. Studies of this kind have shown noteworthy results in cardiovascular diseases, such as coronary artery disease [12, 13], acute coronary syndrome [14], and myocardial infarction [15]. Disease-related knowledge has been found to affect treatment compliance in CHD management, including control of glucose, lipid metabolism indicators, and cardiac rehabilitation [16, 17]. The previous study of CHD undertaken in China reported acceptable attitudes and knowledge regarding medication adherence but a number of barriers to better practice [18]. Although educational interventions based on the KAP model reportedly positively influence the quality of life [19], the efficacy of patient education on changing behavior outcomes for CHD patients remains unclear [20]. Moreover, the prognosis of patients undergoing unplanned secondary PCI is still poor in China, with medical care focusing on therapy and shorter follow-up of treated patients [13, 19]. Consequently, in the present study, we aimed to explore CHD patients' knowledge, attitude, and practice toward CHD secondary prevention and risk factors related to the post-intervention prognosis.

## Methods

### Study design and participants

This web-based cross-sectional study enrolled patients with CHD who visited at the Yangpu District Central Hospital in Shanghai, China, between October 18, 2022, and March 25, 2023. Study participants were CHD patients with at least one vessel stenosis of $\geq$ 50% confirmed by coronary computed tomography angiography (CTA) or coronary angiography. Patients who could not complete the questionnaire, including those with poor writing ability, were excluded.

The study was approved by the Tongji University Affiliated Yangpu Hospital Ethics Committee (Ethics Approval No. LL-2022-SCI-007) and informed consent was obtained from all participants. This observational cross-sectional study complies with the STROBE guidelines.

### Questionnaire and quality control

The questionnaire design was based on previous KAP studies undertaken among CHD patients in general [21] as well as post-PCI CHD patients [22], and was pilot tested on a small scale (30 questionnaires), achieving a reliability of 0.858.

The final questionnaire was in Chinese and included information divided into four categories. The general information section included 14 items. The knowledge section included 21 items, scoring 2–0 assigned for understanding, comparative understanding, and no understanding, respectively. True/false questions were assigned 1 for correct answers and 0 for incorrect or unclear answers. The attitude section included 12 items, scored on a five-point Likert scale ranging from "a" very positive (5 points) to "e" very negative (1 point), with a final score ranging from 12–60 points. The practice section included 12 items, with descriptive analysis used for questions related to the medical knowledge of CHD and a five-point Likert scale used for the remaining questions, with a final score ranging from 11–55.

The questionnaire was designed on Wenjuanxing, a professional online questionnaire software platform provided by Changsha Ranxing Information Technology (China). A link to the questionnaire/paper questionnaire was generated and distributed to potential participants through WeChat groups, outpatient visits and telephone inquiry. Questionnaires with obvious logical errors, two or more unanswered questions, or the pattern of choosing the same option of all KAP items were considered invalid.

### Statistical analysis

All data were analyzed using SPSS 26.0 statistical software (IBM, Armonk, NY, USA). Continuous variables were expressed as mean±SD, and were compared between groups using Kruskal-Wallis H test. Categorical variables were described using frequency (percentage). Logistic regression analysis was used to identify factors associated with good practice. Variables with a P-value less than 0.05 in the univariate analysis were included in the multivariate analysis. Good practice was defined as >70% of a maximum score [23]. The two-sided P<0.05 was considered statistically significant.

## Results

A total of 507 participants were included in the study. Of these, 361 (71.2%) were male, and 146 (28.8%) were female. Participants' ages ranged widely, with 134 individuals (26.4%) aged $\leq$65 years and 373 (73.6%) aged >65 years. In terms of education, 125 (24.7%) had junior high school education or below, 273 (53.8%) completed high school or technical school, and 109

(21.5%) held a college degree or higher. The mean scores for knowledge, attitudes, and practices were 31.28 ± 7.30, 54.09 ± 3.33, and 35.48 ± 3.36, respectively (Table 1).

Significant differences in Knowledge scores were observed across age, education level, job type, monthly income, smoking status, and drinking status. Participants aged ≤65 years had higher Knowledge scores than those aged >65 years (34.67 ± 6.50 vs. 30.06 ± 7.20, P < 0.001).

**Table 1. Baseline characteristics and knowledge, attitude and practice scores of patients.**

| Variables | N (%) | Knowledge, mean ± SD | P | Attitude, mean ± SD | P | Practice, mean ± SD | P |
|---|---|---|---|---|---|---|---|
| **Total scores** | 507(100.0) | 31.28 (7.30) | | 54.09 (3.33) | | 35.48 (3.36) | |
| **Gender** | | | 0.808 | | 0.598 | | 0.102 |
| Male | 361(71.2) | 31.24 (7.34) | | 54.03 (3.35) | | 35.30 (3.44) | |
| Female | 146(28.8) | 31.38 (7.25) | | 54.25 (3.28) | | 35.92 (3.12) | |
| **Age** | | | <0.001 | | <0.001 | | 0.404 |
| ≤65 years | 134(26.4) | 34.67 (6.50) | | 55.72 (3.26) | | 35.22 (3.99) | |
| >65 years | 373(73.6) | 30.06 (7.20) | | 53.51 (3.16) | | 35.57 (3.11) | |
| **Residential area** | | | 0.887 | | 0.629 | | 0.588 |
| Non-urban | 494(97.4) | 31.28 (7.34) | | 54.11 (3.31) | | 35.46 (3.36) | |
| Urban | 13(2.6) | 31.38 (5.90) | | 53.62 (4.11) | | 36.23 (3.39) | |
| **Education** | | | <0.001 | | <0.001 | | 0.528 |
| Junior high school or below | 125(24.7) | 26.02 (7.54) | | 52.62 (2.74) | | 35.70 (3.01) | |
| High school/technical school | 273(53.8) | 32.41 (6.58) | | 53.42 (3.01) | | 35.53 (3.37) | |
| College/bachelor's degree or above | 109(21.5) | 34.47 (5.48) | | 57.46 (2.33) | | 35.08 (3.70) | |
| **Job type** | | | <0.001 | | <0.001 | | 0.914 |
| Formal Employee/Occupation | 67(13.2) | 35.06 (6.18) | | 56.18 (2.83) | | 35.28 (3.88) | |
| Retired | 412(81.3) | 30.53 (7.21) | | 53.69 (3.26) | | 35.52 (3.17) | |
| Other | 28(5.5) | 33.21 (8.17) | | 55.00 (3.46) | | 35.29 (4.68) | |
| **Monthly per capita income, RMB** | | | <0.001 | | <0.001 | | 0.248 |
| <5000 | 362(71.4) | 30.44 (7.21) | | 53.72 (3.22) | | 35.60 (3.34) | |
| ≥5000 | 145(28.6) | 33.39 (7.14) | | 55.04 (3.40) | | 35.18 (3.40) | |
| **Marital status** | | | 0.434 | | 0.340 | | 0.331 |
| Unmarried, divorced or widowed | 473(93.3) | 31.38 (7.18) | | 54.13 (3.35) | | 35.44 (3.38) | |
| Married | 34(6.7) | 29.91 (8.93) | | 53.59 (3.06) | | 36.00 (3.01) | |
| **Smoking status** | | | <0.001 | | 0.166 | | <0.001 |
| Never smoker | 184(36.3) | 31.48 (7.20) | | 54.03 (3.35) | | 36.05 (3.05) | |
| Former smoker | 144(28.4) | 28.78 (7.49) | | 53.74 (3.17) | | 36.02 (3.08) | |
| Current smoker | 179(35.3) | 33.08 (6.70) | | 54.44 (3.41) | | 34.46 (3.64) | |
| **Drinking status** | | | 0.012 | | 0.649 | | <0.001 |
| Never drinker | 383(75.5) | 30.96 (7.41) | | 54.03 (3.38) | | 35.94 (3.17) | |
| Former drinker | 86(17.0) | 31.29 (6.71) | | 54.15 (2.98) | | 34.43 (3.19) | |
| Current drinker | 38(7.5) | 34.47 (6.92) | | 54.63 (3.51) | | 33.18 (4.16) | |
| **Comorbidities: Hypertension** | | | 0.124 | | 0.440 | | 0.211 |
| None | 38(7.5) | 32.13 (9.15) | | 54.53 (3.27) | | 34.74 (4.00) | |
| Yes | 469(92.5) | 31.21 (7.14) | | 54.06 (3.33) | | 35.54 (3.30) | |
| **Comorbidities: Diabetes** | | | 0.962 | | 0.121 | | <0.001 |
| None | 416(82.1) | 31.30 (7.27) | | 54.19 (3.39) | | 35.10 (3.23) | |
| Yes | 91(17.9) | 31.18 (7.51) | | 53.68 (2.98) | | 37.22 (3.39) | |
| **PCI times** | | | 0.339 | | 0.801 | | 0.826 |
| 1 | 461(90.9) | 31.16 (7.40) | | 54.08 (3.33) | | 35.49 (3.37) | |
| 2–3 | 46(9.1) | 32.50 (6.18) | | 54.22 (3.29) | | 35.35 (3.31) | |

Those with college education or higher scored higher in Knowledge than those with high school education or below (34.47 ± 5.48 for college and 26.02 ± 7.54 for junior high school or below, P < 0.001). Formal employees had higher Knowledge scores compared to retired individuals (35.06 ± 6.18 vs. 30.53 ± 7.21, P < 0.001). Monthly per capita income ≥5000 RMB was associated with higher Knowledge scores than income <5000 RMB (33.39 ± 7.14 vs. 30.44 ± 7.21, P < 0.001). Current smokers had higher Knowledge scores than never smokers (33.08 ± 6.70 vs. 31.48 ± 7.20, P < 0.001), and current drinkers scored higher than non-drinkers (34.47 ± 6.92 vs. 30.96 ± 7.41, P = 0.012). In Attitude scores, significant differences were observed based on age, education, job type, and income. Participants aged ≤65 years had higher Attitude scores than those aged >65 years (55.72 ± 3.26 vs. 53.51 ± 3.16, P < 0.001). Higher educational attainment correlated with higher Attitude scores, with college graduates scoring 57.46 ± 2.33 compared to 52.62 ± 2.74 for those with junior high school education or below (P < 0.001). Formal employees had higher Attitude scores than retirees (56.18 ± 2.83 vs. 53.69 ± 3.26, P < 0.001). Monthly income ≥5000 RMB was also associated with higher Attitude scores than income <5000 RMB (55.04 ± 3.40 vs. 53.72 ± 3.22, P < 0.001). Regarding Practice scores, significant differences were observed only for smoking, drinking, and diabetes status. Never smokers had higher Practice scores than current smokers (36.05 ± 3.05 vs. 34.46 ± 3.64, P < 0.001). Non-drinkers scored higher in Practice than current drinkers (35.94 ± 3.17 vs. 33.18 ± 4.16, P < 0.001). Those with diabetes had higher Practice scores than those without (37.22 ± 3.39 vs. 35.10 ± 3.23, P < 0.001) (Table 1).

In terms of knowledge about CHD, participants showed varied levels of familiarity with symptoms and risk factors, as well as understanding of specific CHD-related knowledge items (Table 2 and Fig 1). Regarding symptoms, 56% of participants reported familiarity with chest pain (angina), while 51.7% recognized chest tightness. Lower familiarity was observed for symptoms such as shortness of breath, pale skin or cold sweat, dizziness, and palpitations, each noted by 39.3% of participants. Confusion as a symptom was slightly more recognized, with 41.4% of participants indicating familiarity.

**Table 2. Knowledge of symptoms and risk factors of coronary heart disease among study population.**

| Item, n (%) | Familiar | Somewhat familiar | Unfamiliar |
|---|---|---|---|
| Symptoms of coronary heart disease | | | |
| Chest pain (angina) | 284(56%) | 139(27.4%) | 84(16.6%) |
| Chest tightness | 262(51.7%) | 149(29.4%) | 96(18.9%) |
| Shortness of breath | 199(39.3%) | 170(33.5%) | 138(27.2%) |
| Pale skin, cold sweat | 199(39.3%) | 170(33.5%) | 138(27.2%) |
| Dizziness | 199(39.3%) | 170(33.5%) | 138(27.2%) |
| Palpitations | 199(39.3%) | 170(33.5%) | 138(27.2%) |
| Confusion | 210(41.4%) | 170(33.5%) | 127(25%) |
| Risk Factors for Coronary Heart Disease | | | |
| Smoking | 394(77.7%) | 102(20.1%) | 11(2.2%) |
| Alcohol consumption | 330(65.1%) | 144(28.4%) | 33(6.5%) |
| Diabetes | 365(72%) | 108(21.3%) | 34(6.7%) |
| High level of low-density lipoprotein cholesterol | 360(71%) | 100(19.7%) | 47(9.3%) |
| Poor control of hypertension (blood pressure >140/90 mmHg) | 353(69.6%) | 114(22.5%) | 40(7.9%) |
| Obesity | 392(77.3%) | 100(19.7%) | 15(3%) |
| Lack of physical activity | 337(66.5%) | 118(23.3%) | 52(10.3%) |
| Family history of coronary heart disease | 343(67.7%) | 112(22.1%) | 52(10.3%) |
| High-salt, high-sugar, high-fat diet | 364(71.8%) | 91(17.9%) | 52(10.3%) |
| High psychological stress | 330(65.1%) | 94(18.5%) | 83(16.4%) |

For risk factors, 77.7% of participants identified smoking as a significant risk factor, with obesity recognized by 77.3%. Other commonly acknowledged risk factors included diabetes (72%), high levels of low-density lipoprotein cholesterol (71%), and a high-salt, high-sugar, high-fat diet (71.8%). Awareness was also notable for family history (67.7%), physical inactivity (66.5%), alcohol consumption (65.1%), and high psychological stress (65.1%). Poor blood pressure control was reported as familiar by 69.6% of participants.

Fig 1 illustrates the correctness rates for four specific knowledge items on CHD. For item K2, which states that women are more susceptible to developing CHD, 57.6% of participants answered correctly, while 42.4% answered incorrectly. Item K4, which indicates that physical labor and emotional excitement can trigger a CHD attack, had a higher correctness rate, with 94.1% answering correctly. Item K5, regarding the necessity of long-term medication and regular follow-up after a CHD diagnosis, had the highest correctness rate, with 98.8% of participants answering correctly. For item K6, which suggests that recurrence of CHD is unlikely after undergoing PCI surgery, 84.6% answered correctly, while 15.4% answered incorrectly.

Participants generally reported strong attitudes towards coronary heart disease prevention. Quitting smoking was viewed as crucial by 96.8%, while 87.6% agreed on the importance of quitting drinking. Controlling blood glucose was rated important by 70.2%, with another 17.6% agreeing. For blood pressure control, 62.5% strongly agreed on its importance, and 58.2% felt similarly about controlling blood lipid levels. Controlling weight received slightly lower agreement, with 50.5% strongly agreeing and 29.8% agreeing. A healthy diet was rated important by all participants, and moderate exercise was viewed positively by 54%. Maintaining a positive mentality and regular follow-up visits were also rated highly, with over 95% agreement across these items (Table 3).

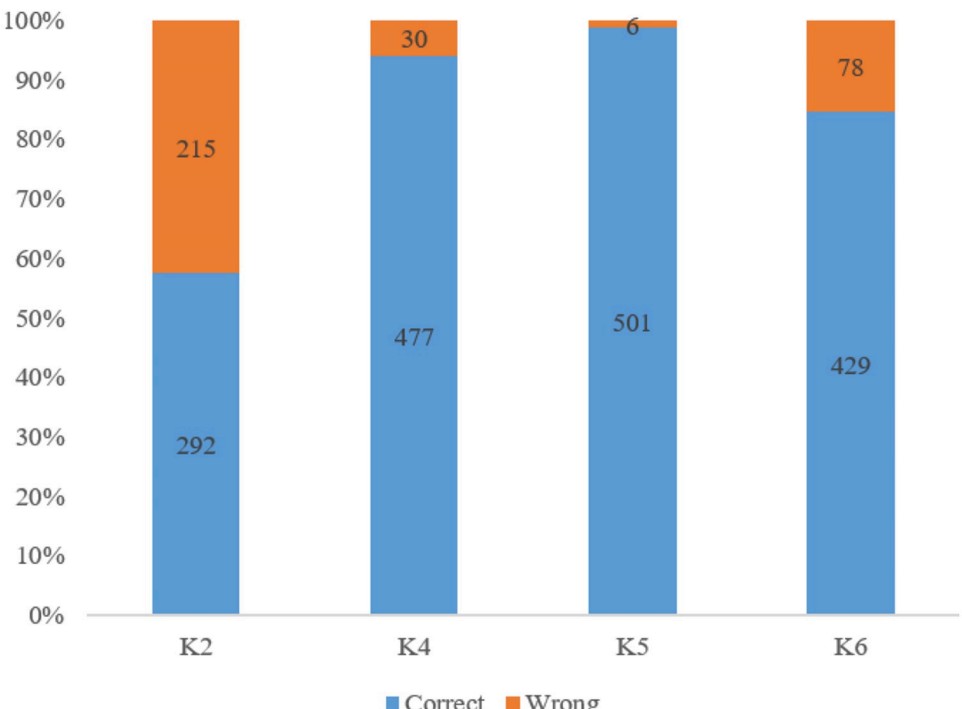

**Fig 1. Correctness rate for chosen items in the knowledge dimension.** K2: Women are more susceptible to developing coronary heart disease. K4: Physical labor and emotional excitement can trigger an attack of coronary heart disease. K5: After being diagnosed with coronary heart disease, long-term regular medication and regular follow-up visits are necessary. K6: After undergoing PCI surgery, the recurrence of coronary heart disease is unlikely.

**Table 3. Attitude towards coronary heart disease secondary preventions.**

| Item, n (%) | Strongly agree | Agree | Neutral | Disagree | Strongly disagree |
|---|---|---|---|---|---|
| Quitting smoking is crucial for controlling coronary heart disease. | 491(96.8%) | 15(3%) | 1(0.2%) | 0 (0%) | 0 (0%) |
| Quitting drinking is crucial for controlling coronary heart disease. | 182(35.9%) | 262(51.7%) | 34(6.7%) | 24(4.7%) | 5(1%) |
| Controlling blood glucose levels is crucial for controlling coronary heart disease. | 356(70.2%) | 89(17.6%) | 30(5.9%) | 26(5.1%) | 6(1.2%) |
| Controlling blood pressure is crucial for controlling coronary heart disease. | 317(62.5%) | 174(34.3%) | 15(3%) | 1(0.2%) | 0 (0%) |
| Controlling blood lipid levels is crucial for controlling coronary heart disease. | 295(58.2%) | 155(30.6%) | 37(7.3%) | 18(3.6%) | 2(0.4%) |
| Controlling weight is crucial for controlling coronary heart disease. | 256(50.5%) | 151(29.8%) | 63(12.4%) | 33(6.5%) | 4(0.8%) |
| Maintaining a healthy diet is crucial for controlling coronary heart disease. | 306(60.4%) | 201(39.6%) | 0 (0%) | 0 (0%) | 0 (0%) |
| Moderate exercise is crucial for controlling coronary heart disease. | 274(54%) | 150(29.6%) | 38(7.5%) | 37(7.3%) | 8(1.6%) |
| Maintaining a positive mentality is crucial for controlling coronary heart disease. | 288(56.8%) | 148(29.2%) | 53(10.5%) | 17(3.4%) | 1(0.2%) |
| Regular medication is crucial for controlling coronary heart disease. | 297(58.6%) | 199(39.3%) | 7(1.4%) | 4(0.8%) | 0 (0%) |
| Regular follow-up visits are crucial for controlling coronary heart disease. | 369(72.8%) | 138(27.2%) | 0 (0%) | 0 (0%) | 0 (0%) |
| Concerned that your coronary heart disease is worsening. | 392(77.3%) | 66(13%) | 41(8.1%) | 8(1.6%) | 0 (0%) |

Medication adherence was high, with 84.8% indicating they took medication on time daily, and 82.6% reported regular hospital visits. Timely medical care in response to discomfort, such as chest pain, was practiced consistently by 50.7%, with an additional 17.8% often doing so. For lifestyle habits, daily smoking was reported by 8.5%, while 72.4% rarely or never smoked. Alcohol consumption was also low, with 3.7% drinking daily and 89.3% rarely or never drinking. Daily blood glucose monitoring was practiced by 4.1%, and daily blood pressure checks by 8.7%. Exercise frequency varied, with 6.5% engaging daily and 27.4% more than three times a week. Adherence to a low-salt, low-sugar, low-fat diet was reported by 7.7%, and 34.7% maintained a positive mentality often, although 39.5% only sometimes or rarely did so (Table 4).

Multivariate analysis indicated that smoking and diabetes status were significantly associated with Practice scores. Current smokers reported lower practice levels than never smokers (OR = 2.858, 95% CI: 1.442–5.662, P = 0.003). Participants with diabetes reported higher practice levels than those without diabetes (OR = 4.169, 95% CI: 2.329–7.463, P < 0.001) (Table 5).

**Table 4. Response status of each item in the practice dimension.**

| Item, n (%) | Always | Often | Sometimes | Rarely | Never |
|---|---|---|---|---|---|
| Take medication on time every day | 430(84.8%) | 46(9.1%) | 26(5.1%) | 5(1%) | 0 (0%) |
| Regularly go to the hospital for medical treatment | 419(82.6%) | 49(9.7%) | 29(5.7%) | 10(2%) | 0 (0%) |
| Seek medical treatment in a timely manner when experiencing discomfort such as chest pain or tightness | 257(50.7%) | 90(17.8%) | 67(13.2%) | 78(15.4%) | 15(3%) |
| | Every day | More than 3 times a week | 1–2 times a week | More than 2 times a month | Almost never |
| Smoking | 43(8.5%) | 29(5.7%) | 21(4.1%) | 47(9.3%) | 367(72.4%) |
| Drinking | 19(3.7%) | 8(1.6%) | 11(2.2%) | 16(3.2%) | 453(89.3%) |
| Measure blood glucose | 21(4.1%) | 22(4.3%) | 13(2.6%) | 42(8.3%) | 409(80.7%) |
| Measure blood pressure | 44(8.7%) | 138(27.2%) | 99(19.5%) | 116(22.9%) | 110(21.7%) |
| Exercise | 33(6.5%) | 139(27.4%) | 118(23.3%) | 124(24.5%) | 93(18.3%) |
| Low-salt, low-sugar, low-fat diet | 39(7.7%) | 158(31.2%) | 108(21.3%) | 132(26%) | 70(13.8%) |
| Maintain a good mentality | 41(8.1%) | 176(34.7%) | 90(17.8%) | 151(29.8%) | 49(9.7%) |

**Table 5. Analysis of factors affecting good practice of coronary heart disease secondary prevention.**

| Practice | Univariate analysis OR(95%CI) | P | Multivariate analysis OR(95%CI) | P |
|---|---|---|---|---|
| **Knowledge** | 1.023 (0.998,1.045) | 0.067 | 0.999 (0.969,1.030) | 0.940 |
| **Attitude** | 0.989 (0.938,1.043) | 0.692 | 0.934 (0.872,1.002) | 0.056 |
| **Gender** | | | | |
| Male | | | | |
| Female | 0.420 (0.283,0.621) | <0.001 | 0.514 (0.264,1.001) | 0.050 |
| **Age** | | | | |
| ≤65 years | | | | |
| >65 years | 0.460 (0.298,0.698) | <0.001 | 0.670 (0.333,1.347) | 0.261 |
| **Residence area** | | | | |
| Urban | | | | |
| Non-urban | 0.623 (0.198,1.903) | 0.402 | | |
| **Education** | | | | |
| Junior high school or below | | | | |
| High school/Technical school | 1.213 (0.792,1.855) | 0.374 | 0.998 (0.590,1.689) | 0.994 |
| College or above | 1.796 (1.062,3.066) | 0.030 | 1.547 (0.699,3.424) | 0.282 |
| **Job type** | | | | |
| Formal employee | | | | |
| Retired | 0.367 (0.197,0.650) | 0.001 | 0.646 (0.269,1.553) | 0.329 |
| Others | 0.662 (0.253,1.795) | 0.406 | 0.779 (0.260,2.340) | 0.657 |
| **Monthly per capita income** | | | | |
| <5000 | | | | |
| ≥5000 | 1.406 (0.949,2.099) | 0.092 | 0.755 (0.453,1.259) | 0.282 |
| **Marital status** | | | | |
| Unmarried, divorced or widowed | 1.204 (0.596,2.522) | 0.611 | | |
| Married | | | | |
| **Smoking status** | | | | |
| Never smoked | | | | |
| Former smoker | 1.120 (0.723,1.735) | 0.613 | 0.803 (0.419,1.542) | 0.510 |
| Current smoker | 4.368 (2.781,6.968) | <0.001 | 2.858 (1.442,5.662) | 0.003 |
| **Drinking status** | | | | |
| Never drinker | | | | |
| Former drinker | 1.780 (1.095,2.950) | 0.022 | 1.352 (0.771,2.371) | 0.293 |
| Current drinker | 2.406 (1.173,5.334) | 0.022 | 1.021 (0.432,2.411) | 0.963 |
| **Comorbidities: Hypertension** | | | | |
| None | | | | |
| Yes | 1.243 (0.635,2.414) | 0.521 | | |
| **Comorbidities: Diabetes** | | | | |
| None | | | | |
| Yes | 3.376 (2.004,5.943) | <0.001 | 4.169 (2.329,7.463) | <0.001 |
| **PCI times** | | | | |
| 1 | | | | |
| 2–3 times | 1.284 (0.694,2.448) | 0.434 | | |

## Discussion

Patients with CHD in Shanghai, China, demonstrated good knowledge and positive attitudes toward CHD secondary prevention, though certain gaps in practice remain evident. Improving adherence to lifestyle modifications, especially for high-risk groups such as smokers and those without diabetes, may enhance the effectiveness of secondary prevention efforts.

The study population predominantly included men aged >65 years, which is close to the ordinary profile of CHD patients in China [24]. However, with global aging, CHD has also become the leading cause of mortality among Chinese women [25]. Currently, there is scarce information on the factors affecting CHD in female populations, which leads to the under-recognition of their cardiovascular risk [25, 26]. In this study, the proportion of female responders was relatively low; although practice scores appeared higher in women, logistic regression analysis did not confirm a significant association. Moreover, in the knowledge dimension, awareness about whether women are more susceptible to developing CHD was among the least recognized items, with a small percentage of participants correctly identifying it. This lack of knowledge could result in missed CHD symptoms due to incorrect expectations.

The mean practice score in this study indicates several barriers to good practice, which aligns with the generally low compliance with cardiac rehabilitation in China [19]. Previous Chinese studies also reported good knowledge/attitudes but lower practice scores regarding adherence to secondary prevention measures in CHD [18, 20] and CAD [13]. In contrast, a study from Lebanon reported low knowledge scores among myocardial infarction patients, while a U.S. study on CAD found poor knowledge and practice that improved significantly following educational interventions. This suggests that the population in the present study may have unique features influencing the relationship between knowledge and practice.

Previously reported barriers to good practice towards CHD secondary prevention included younger age and chronic diseases [18, 21], education level [15], ethnicity, gender, and exposure to health information in the media [14, 22]. In this study, contrary to the results reported by Yu *et al.* [18], diabetes mellitus was associated with higher practice scores. Although the study design limitations prevent direct comparison, it is notable that diabetes is frequently targeted by in-hospital education programs. Some dietary and prevention measures overlap with CHD prevention, potentially receiving more attention. On the other hand, a concerning finding was that a portion of participants rarely measured their blood pressure, despite hypertension being the most common comorbidity. Even some CHD patients diagnosed with hypertension measured their blood pressure infrequently, suggesting that existing educational programs may not sufficiently emphasize blood pressure control's importance in CHD.

Higher levels of physical activity and lifestyle correction have been associated with a lower risk of CHD and are considered essential to secondary prevention [27, 28]. However, this study's attitude assessment revealed that the statement, "Moderate exercise is crucial for controlling CHD," was among the more controversial, with a small percentage of participants disagreeing. Additionally, some participants reported rarely engaging in exercise, and others exercised only occasionally. These findings are consistent with previous studies, despite variations in knowledge and attitude [14, 15]. Furthermore, a significant portion of the participants were current smokers, but only a small percentage admitted to being current drinkers. Attitude assessment showed that a comparatively lower proportion strongly agreed that quitting drinking is crucial for controlling CHD (whereas nearly all strongly agreed on the importance of quitting smoking), suggesting that drinking may be underreported and may more significantly impact practice, as confirmed by the multivariate logistic regression showing that drinking habits (both former and current) were significantly associated with lower practice scores.

Improving adherence to CHD secondary prevention measures remains necessary among the study population. Although health literacy and disease knowledge play a critical role in improving practice outcomes, knowledge gained from educational interventions does not always predict improvements in attitudes or exercise participation [12, 13], consistent with the weak associations between knowledge scores and both practice and attitude, and the insignificant association between attitude and practice observed in this study. The findings further confirm that knowledge had a direct effect on attitude but not on practice, and that attitude did not significantly influence practice. Alternative models for delivering knowledge are needed, beyond in-hospital education, to address these gaps. Early reports suggest that using one-way text messages for cardiac rehabilitation and lifestyle management without imposing additional pressure on patients may be promising [6, 11, 29]. Future studies should explore whether these interventions can save time and resources and effectively prevent CHD recurrence.

## Limitation of this study

Firstly, this was a single-center study, which might lead to selection bias. Moreover, excluding participants who could not write contributed to this selection bias. Secondly, the sample size might not be enough to reveal significant differences between study sub-populations. In addition, male and female participants were analyzed together. Finally, responders might have opted for socially acceptable answers instead of truth (especially regarding smoking and drinking habits), which might lead to additional bias.

## Conclusions

In conclusion, patients with CHD in Shanghai, China, demonstrated good knowledge and positive attitudes toward CHD secondary prevention, although certain areas of practice showed room for improvement. Clinically, targeted educational interventions focusing on improving specific preventive practices, particularly for smokers and those without diabetes, may enhance overall adherence to CHD secondary prevention measures.

## Supporting information

**S1 File. Questionnaire-English.**
(DOCX)

**S2 File. Original data table-English.**
(XLSX)

**S1 Checklist. STROBE-checklist.**
(DOCX)

## Author Contributions

**Data curation:** Hao Wang, Wenqi Guan, Tan Zhou, Hongbao Wang, Wei Li, Xueqin He.

**Formal analysis:** Hao Wang, Bo Wu, Wei Li.

**Methodology:** Bo Wu, Wenqi Guan, Hongbao Wang.

**Project administration:** Tan Zhou.

**Resources:** Xueqin He.

**Supervision:** Bo Wu, Wenqi Guan.

**Visualization:** Wenqi Guan, Xueqin He.

**Writing – original draft:** Hao Wang.

**Writing – review & editing:** Hao Wang.

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
