## [Decision Letter · Decision Letter 0]

18 Sep 2024

PONE-D-24-36121Knowledge, Attitude, and Practice toward Coronary Heart Disease Secondary Prevention among Coronary Heart Disease Patients in Shanghai, ChinaPLOS ONE

Dear Dr. Wang,

Thank you for submitting your manuscript to PLOS ONE. After careful consideration, we feel that it has merit but does not fully meet PLOS ONE’s publication criteria as it currently stands. Therefore, we invite you to submit a revised version of the manuscript that addresses the points raised during the review process.

**Please make the following minor revisions to your paper:-**

**1. As this is an observational cross-sectional study, please ensure that it is compliant with STROBE guidelines** (https://www.strobe-statement.org) **and mention it in the methodology.**

**2. Please address the points raised by the 2 reviewers.**

We look forward to receiving your revised manuscript.

Kind regards,

Hean Teik Ong

Academic Editor

PLOS ONE

**Journal Requirements:**

4. We note that you have referenced "Dugani SB, Moorthy MV, Li C, Demler OV, Alsheikh-Ali AA, Ridker PM, et al." which has currently not yet been accepted for publication. Please remove this from your References and amend this to state in the body of your manuscript: (Dugani SB, Moorthy MV, Li C, Demler OV, Alsheikh-Ali AA, Ridker PM, et al. [Submitted]) as detailed online in our guide for authors

**Additional Editor Comments:**

Thank you for your submission.

Please make the following minor revisions to your paper:-

1. As this is an observational cross-sectional study, please ensure that it is compliant with STROBE guidelines (https://www.strobe-statement.org) and mention it in the methodology.

2. Please address the points raised by the 2 reviewers.

Reviewers' comments:

Reviewer's Responses to Questions

**Comments to the Author**

1. Is the manuscript technically sound, and do the data support the conclusions?

Reviewer #1: Yes

Reviewer #2: Yes

2. Has the statistical analysis been performed appropriately and rigorously? 

Reviewer #1: Yes

Reviewer #2: I Don't Know

3. Have the authors made all data underlying the findings in their manuscript fully available?

Reviewer #1: Yes

Reviewer #2: Yes

4. Is the manuscript presented in an intelligible fashion and written in standard English?

Reviewer #1: Yes

Reviewer #2: Yes

5. Review Comments to the Author

**Reviewer #1:** Just a few comments:

Line 33: To Make the meaning clearer the sentence in Conclusion section should be written as —- Good Knowledge and presence of diabetes mellitus were associated with better practice whereas smoking and drinking habits were associated with the poor practice.

Similar changes applied to conclusion section in line 234-235

Line 47– should be low density lipoprotein

Before line 225: insert subtitle “Limitation of this study”

Exclude those who cannot write — selection bias — is one of the limitations of this study too.

Line 236— What is the meaning of “Discussed gaps” here?

**Reviewer #2:** Some comments about the introduction:

Line 40: shouldn’t the word be ‘atherosclerosis’ rather than ‘arteriosclerosis’?

Line 46-49: Main risk factors for CHD did not include commonly recognized factors like: LDL cholesterol, hypertension and smoking.

Line 121-122: majority of the population studied (95%) had PCI. Perhaps should focus on this group alone. Why was the 5% on non PCI patients included, in the population studied?

Table 3 Attitude toward Coronary Heart disease secondary prevention

Last 2 lines have the same statement: 'Regular follow up visits are crucial for controlling coronary heart disease'. But the number of response are difference. Which are the correct values?

6. PLOS authors have the option to publish the peer review history of their article (what does this mean?). If published, this will include your full peer review and any attached files.

Reviewer #1: No

Reviewer #2: No

---

## [Author Response · Author response to Decision Letter 0]

2 Dec 2024

Dear Hean Teik Ong,

Thank you for the valuable feedback on our manuscript titled "Knowledge, Attitude, and Practice toward Coronary Heart Disease Secondary Prevention among Coronary Heart Disease Patients in Shanghai, China". I have carefully considered the reviewers' comments and made comprehensive revisions accordingly. Attached to this letter, you will find the revised manuscript along with a point-by-point response to each comment. I hope these revisions adequately address the reviewers' concerns and make our paper a suitable candidate for publication in PLOS ONE.

Sincerely,

Hao Wang

E-mail: dr_wanghao@163.com

Tel: 86-18916538685

Point-by-Point Response

Journal Requirements

Comment 1: 

We note that you have referenced "Dugani SB, Moorthy MV, Li C, Demler OV, Alsheikh-Ali AA, Ridker PM, et al." which has currently not yet been accepted for publication. Please remove this from your References and amend this to state in the body of your manuscript: (Dugani SB, Moorthy MV, Li C, Demler OV, Alsheikh-Ali AA, Ridker PM, et al. [Submitted]) as detailed online in our guide for authors

Response:

Thank you for pointing out this issue. We have now revised the reference.

Additional Editor Comments

Comment 1: 

Please make the following minor revisions to your paper:-

As this is an observational cross-sectional study, please ensure that it is compliant with STROBE guidelines (https://www.strobe-statement.org) and mention it in the methodology.

Response:

Thank you for your feedback. We have ensured that the study complies with the STROBE guidelines and have mentioned this in the methodology section as requested.

Comment 2: 

Please address the points raised by the 2 reviewers.

Response:

Thank you for your guidance. We have carefully addressed all the points raised by the two reviewers.

Reviewer #1: Just a few comments:

Comment 1: 

Line 33: To Make the meaning clearer the sentence in Conclusion section should be written as —- Good Knowledge and presence of diabetes mellitus were associated with better practice whereas smoking and drinking habits were associated with the poor practice.

Similar changes applied to conclusion section in line 234-235

Response:

Thank you for your thoughtful suggestion. We have revised both the abstract and the conclusion section to improve clarity, aligning with your recommended phrasing.

Comment 2: 

Line 47– should be low density lipoprotein

Response:

Thank you, it has been revised.

Comment 3: 

Before line 225: insert subtitle “Limitation of this study”

Response:

Thank you, the suggested subtitle "Limitation of this study" has been added.

Comment 4: 

Exclude those who cannot write — selection bias — is one of the limitations of this study too.

Response:

Thank you, the limitations section has been revised to include this aspect of selection bias.

Comment 5: 

Line 236— What is the meaning of “Discussed gaps” here?

Response:

Thank you for your suggestion. We have replaced "discussed gaps" with "identified knowledge gaps and barriers" in the conclusion to enhance clarity.

Reviewer #2: Some comments about the introduction:

Comment 1: 

Line 40: shouldn’t the word be ‘atherosclerosis’ rather than ‘arteriosclerosis’?

Response:

Thank you, the word has been corrected to 'atherosclerosis.'

Comment 2: 

Line 46-49: Main risk factors for CHD did not include commonly recognized factors like: LDL cholesterol, hypertension and smoking.

Response:

Thank you, it has been revised.

Comment 3: 

Line 121-122: majority of the population studied (95%) had PCI. Perhaps should focus on this group alone. Why was the 5% on non PCI patients included, in the population studied?

Response:

Thank you for your insightful comment. We have completely reanalyzed the study, focusing solely on the 95% of patients who had PCI. The results and discussion have been updated based on this revised analysis.

Comment 4: 

Table 3 Attitude toward Coronary Heart disease secondary prevention

Response:

Thank you for your feedback. We have revised Table 3.

Comment 5: 

Last 2 lines have the same statement: 'Regular follow up visits are crucial for controlling coronary heart disease'. But the number of response are difference. Which are the correct values?

Response:

We apologize for the confusion. In Table 3, the result in the second-to-last row represents responses to the statement “Regular follow-up visits are crucial for controlling coronary heart disease,” which corresponds to the original Chinese question: “您认为定期复查对控制冠心病病情非常重要。” The last row represents responses to the question “Concerned that your coronary heart disease is worsening,” translated from the Chinese: “您担心您的冠心病病情恶化。” The values have been corrected accordingly.

---

## [Editor Report · Decision Letter 1]

4 Dec 2024

Knowledge, Attitude, and Practice toward Coronary Heart Disease Secondary Prevention among Coronary Heart Disease Patients in Shanghai, China

PONE-D-24-36121R1

Dear Dr. Hao Wang,

We’re pleased to inform you that your manuscript has been judged scientifically suitable for publication and will be formally accepted for publication once it meets all outstanding technical requirements.

Kind regards,

Hean Teik Ong

Academic Editor

PLOS ONE
---

## [Editor Report · Acceptance letter]

10 Jan 2025

PONE-D-24-36121R1 

PLOS ONE

Dear Dr. Wang, 

I'm pleased to inform you that your manuscript has been deemed suitable for publication in PLOS ONE. Congratulations! Your manuscript is now being handed over to our production team.

Kind regards, 

on behalf of

Dr. Hean Teik Ong 

Academic Editor

PLOS ONE